# The effects of an evidence- and theory-informed feedback intervention on opioid prescribing for non-cancer pain in primary care: A controlled interrupted time series analysis

**Sarah L. Alderson**[1]*, **Tracey M. Farragher**[2], **Thomas A. Willis**[1], **Paul Carder**[3], **Stella Johnson**[3], **Robbie Foy**[1]

1 Leeds Institute of Health Science, University of Leeds, Leeds, United Kingdom, 2 Division of Population Health, Health Services Research and Primary Care, University of Manchester, Manchester, United Kingdom, 3 West Yorkshire Research and Development, National Health Service Bradford Districts Clinical Commissioning Group, Bradford, United Kingdom

* s.l.alderson@leeds.ac.uk

**Data Availability Statement:** Data cannot be shared publicly because of risk of patient identification where small numbers of patients per

## Abstract

### Background

The rise in opioid prescribing in primary care represents a significant international public health challenge, associated with increased psychosocial problems, hospitalisations, and mortality. We evaluated the effects of a comparative feedback intervention with persuasive messaging and action planning on opioid prescribing in primary care.

### Methods and findings

A quasi-experimental controlled interrupted time series analysis used anonymised, aggregated practice data from electronic health records and prescribing data from publicly available sources. The study included 316 intervention and 130 control primary care practices in the Yorkshire and Humber region, UK, serving 2.2 million and 1 million residents, respectively. We observed the number of adult patients prescribed opioid medication by practice between July 2013 and December 2017. We excluded adults with coded cancer or drug dependency. The intervention, the Campaign to Reduce Opioid Prescribing (CROP), entailed bimonthly, comparative, and practice-individualised feedback reports to practices, with persuasive messaging and suggested actions over 1 year. Outcomes comprised the number of adults per 1,000 adults per month prescribed any opioid (main outcome), prescribed strong opioids, prescribed opioids in high-risk groups, prescribed other analgesics, and referred to musculoskeletal services. The number of adults prescribed any opioid rose pre-intervention in both intervention and control practices, by 0.18 (95% CI 0.11, 0.25) and 0.36 (95% CI 0.27, 0.46) per 1,000 adults per month, respectively. During the intervention period, prescribing per 1,000 adults fell in intervention practices (change −0.11; 95% CI −0.30, −0.08) and continued rising in control practices (change 0.54; 95% CI 0.29, 0.78),

practice are included. Data are available from the University of Leeds School of Medicine Ethics Committee (contact via fmhuniethics@leeds.ac.uk) for researchers who meet the criteria for access to confidential data.

**Funding:** SA received a starter grant for clinical lecturers from the Academy of Medical Sciences, The Wellcome Trust, Medical Research Council, British Heart Foundation, Arthritis Research UK, the Royal College of Physicians and Diabetes UK [Grant number: SGL017/1033] to undertake this research. URL: https://acmedsci.ac.uk/grants-and-schemes/grant-schemes/starter-grants The funders had no role in the study design, data collection and analysis, decision to publish or preparation of the manuscript.

**Competing interests:** The authors have declared that no competing interests exist.

**Abbreviations:** CCG, clinical commissioning group; EHR, electronic health record; IMD, Index of Multiple Deprivation; ITS, interrupted time series; LMM, linear mixed-effects model; NSAID, non-steroidal anti-inflammatory drug.

with a difference of −0.65 per 1,000 patients (95% CI −0.96, −0.34), corresponding to 15,000 fewer patients prescribed opioids. These trends continued post-intervention, although at slower rates. Prescribing of strong opioids, total opioid prescriptions, and prescribing in high-risk patient groups also generally fell. Prescribing of other analgesics fell whilst musculoskeletal referrals did not rise. Effects were attenuated after feedback ceased. Study limitations include being limited to 1 region in the UK, possible coding errors in routine data, being unable to fully account for concurrent interventions, and uncertainties over how general practices actually used the feedback reports and whether reductions in prescribing were always clinically appropriate.

## Conclusions

Repeated comparative feedback offers a promising and relatively efficient population-level approach to reduce opioid prescribing in primary care, including prescribing of strong opioids and prescribing in high-risk patient groups. Such feedback may also prompt clinicians to reconsider prescribing other medicines associated with chronic pain, without causing a rise in referrals to musculoskeletal clinics. Feedback may need to be sustained for maximum effect.

## Author summary

### Why was this study done?

- Opioid prescribing for non-cancer pain is rising despite limited knowledge on effectiveness and increasing evidence of harms, such as falls, fractures, overdose, and addiction.

- There are large differences in opioid prescribing between practices, suggesting prescribing is driven by clinician habits rather than patient need.

- We delivered evidence-based and theory-informed feedback reports to 316 general practices in Yorkshire, UK, every 2 months for 1 year, intended to reduce opioid prescribing by prompting physicians to think twice before starting patients on opioid medication and to review patients not currently benefiting from the medication.

### What did the researchers do and find?

- We looked at trends in the number of patients prescribed opioids for non-cancer pain before, during, and after the intervention in 316 practices that received the feedback compared to 130 practices that did not.

- We also assessed changes in prescribing in patients at higher risk of longer or stronger opioid prescribing, and changes in the prescribing of other painkiller medications, to look at wider impacts on prescribing for pain.

- During the intervention period, the number of adults prescribed any opioid per 1,000 patients per month fell in intervention practices (change −0.11; 95% CI −0.30, −0.08) and rose in control practices (change 0.54; 95% CI 0.29, 0.78), with a difference of −0.65

(95% CI −0.96, −0.34), corresponding to 15,000 fewer patients prescribed opioids at the end of the intervention year.

- Prescribing of strong opioids, total opioid prescriptions, and prescribing in high-risk groups generally fell, although effects lessened after the feedback stopped.

- Prescribing of painkillers not specifically targeted by feedback also fell, without any increases in referrals to musculoskeletal services.

### What do these findings mean?

- Repeated comparative feedback offers a promising and relatively efficient population-level approach to reduce opioid prescribing in primary care.

- Feedback may need to be sustained for maximum effect.

## Introduction

Opioid prescribing is an internationally recognised threat to population health and a pressing challenge for healthcare services [1–5]. Prescription opioid use in the US has fallen little from 2010 peaks, despite increased awareness of risks and opioid abuse [6]. North America is experiencing an 'opioid crisis', with rapidly rising opioid-related mortality, initially due to prescription opioids and more recently due to illicit heroin and fentanyl use, reaching a peak in 2016 [7]. Other higher-income countries risk following similar trajectories [8]. These trends are largely attributed to prescribing for chronic non-cancer pain [9], where opioids are no more effective than non-opioid pain medications and are associated with increased falls, fractures, dependence, overdose, and mortality [10,11]. Despite increased awareness of the potential harms in opioid prescribing, prescription rates remain historically high in both North America and Europe [12–14].

Whilst a growing body of work has investigated problematic opioid prescribing [1,12,15–17], less attention has been paid to evaluating proposed solutions. A Cochrane review found inadequate evidence for interventions targeting opioid use in individuals with chronic pain [18]. However, more recent studies indicate the value of provider- and system-level interventions [19–21], including a multifaceted approach comprising nurse care management, an electronic registry, data-driven academic detailing, and electronic decision tools [20].

Observed large variations in opioid prescribing, up to 10-fold in a UK study of primary care practices, suggests that physician habits and norms are a major driver, rather than patient need and evidence of benefit [16]. An 'upstream' population approach would therefore aim to change physician behaviour around both initiating and continuing opioid prescribing. The audit and feedback approach involves giving healthcare providers a summary of their clinical performance over a specified period [22]. It generally has modest effects on healthcare practice, which can translate into substantial population impacts [22].

We devised and applied an evidence- and theory-informed feedback intervention, the Campaign to Reduce Opioid Prescribing (CROP), to reduce opioid prescribing in primary care by prompting physicians to initiate opioids with caution and review patients currently prescribed opioids with no clear individual benefit. We evaluated the effect of the feedback intervention

on prescribing of opioids and, anticipating the possibility of unintended consequences, prescribing of other analgesics and referrals to musculoskeletal services.

## Methods

### Study design and setting

In the UK, primary care is provided by general practices. Contracts for providing medical care relate to the practice rather than individual physicians. Patients are registered with a single practice rather than individual general practitioners, with an average practice list size of 9,000 patients and a single common electronic health record (EHR). The Yorkshire and Humber region covers an ethnically diverse population of 5.4 million residents with above average socioeconomic deprivation levels [23,24]. This study was conceived from our previous work that showed a rise in opioid prescribing in Leeds and Bradford, the 2 largest cities in West Yorkshire. Medicines optimisation leads, employed by clinical commissioning groups (CCGs), for West Yorkshire asked us to deliver an intervention to reduce opioid prescribing in this area. West Yorkshire (intervention group) has a population of 2.2 million residents served by 317 practices organised within 10 CCGs in 2016. One practice declined data sharing for this study. Five CCGs from the wider Yorkshire and Humber region (outside of West Yorkshire), with a population of 1 million residents and 130 practices, provided control data. We chose to use CCGs in the same region as our intervention sample for the control sample, as these would be subject to similar region-wide prescribing initiatives. Our main study population, and hence sample size, was therefore limited by the coverage of the data-sharing agreements. A further 3 CCGs in the region comprising 134 practices and approximately 650,000 residents were included as additional controls in an analysis using publicly available prescribing data.

We conducted a controlled interrupted time series (ITS) analysis. Controlled ITS is a quasi-experimental design used to evaluate the longitudinal effects of interventions, through regression modelling. The addition of a control group minimises potential confounding from concurrent interventions [25]. This design can detect whether an intervention effect is significantly greater than underlying trends and is appropriate in evaluating area-wide service improvement strategies when randomisation is not feasible [26,27].

### Intervention

Evidence- and theory-informed feedback [28] to each practice reported the number of patients 18 years and older prescribed opioids in the preceding 8 weeks, excluding those with coded cancer, palliative care, or drug dependence, compared to other practices within their CCG and West Yorkshire, as well as changes over time. We did not define clinical categories, given highly variable diagnostic coding for painful conditions. Report content and formats followed a design previously demonstrated to reduce high-risk prescribing in primary care that addressed identified Theoretical Domains Framework determinants of adherence to quality indicators [28,29]. Reports emphasised 'thinking twice' before initiating opioids, rather than addressing the more complex patients prescribed multiple opioids (see S1 Text for the TIDieR summary and S2 Text for an illustrative report). Feedback highlighted patient groups at higher risk of long-term or stronger opioid prescribing: for example, individuals 75 years and older, individuals with coded mental health diagnoses, and individuals co-prescribed antidepressants [16]. The reports incorporated evidence-informed behaviour change techniques, such as specific recommendations for action and action plans, designed to enhance effectiveness [30]. Given the competing priorities and demands that primary care physicians face in routine practice, the reports used non-judgmental and encouraging language. We granted practices access to our EHR searches, allowing them to identify and review individual patients.

Practices received a total of 6 bimonthly reports. We posted 5 copies of each report to practice managers, and the local medicine optimisation leads emailed PDF copies to practice managers for 8 out of 10 CCGs.

The intervention did not involve any changes to existing musculoskeletal or pain services, which general practices and patients could access as usual throughout the study period.

## Data sources and outcomes

Our primary outcome was the number of adults prescribed any opioid per 1,000 adults per month. Secondary outcomes included the number of adults prescribed any opioid per 1,000 adults per month in the high-risk groups highlighted in the feedback [16]. Co-prescription with antidepressants was used as a proxy for mental health illness to reflect our previous work that identified that mental health diagnoses are often poorly recorded in EHRs in the UK [31]. We collected retrospective aggregated, anonymised practice-level data for intervention and control CCGs through the centralised reporting of 2 EHR systems (The Phoenix Partnership SystmOne and EMIS Health), at monthly intervals for 3 periods: pre-intervention (1 July 2013 to 31 March 2016; 47 months), intervention (1 April 2016 to 31 March 2017; 12 months), and post-intervention (1 April 2017 to 31 December 2017; 9 months). We extracted data on numbers of adults prescribed opioids in the previous 8 weeks, excluding those ever coded with cancer, palliative care, or drug dependence. We categorised opioid strength according to World Health Organization reported potency [16]. 'Weaker' opioids (with or without acetaminophen or ibuprofen) comprised codeine, dihydrocodeine, tramadol, pethidine, meptazinol, and tapentadol. 'Strong' opioids comprised diamorphine, morphine, oxycodone, fentanyl, hydromorphone, buprenorphine (excluding preparations used for substance misuse), pentazocine, dipipanone, and papaveretum. We collected data to assess any potential wider impacts on prescribing for pain, specifically the number of adults prescribed non-steroidal anti-inflammatory drugs (NSAIDs) and gabapentinoids, and referrals to musculoskeletal services (see S3 Text for sample search). We converted the numbers of adults in a prescription category into monthly rates based on monthly numbers of relevant adults per practice. The denominator for all outcomes was the number of adults per practice per month, except for the number of adults aged over 75 years prescribed opioids, where the number of adults aged over 75 years per practice per month was used. No patient-level data were extracted to calculate morphine equivalent doses.

We collected monthly data on total opioid prescriptions from the publicly available Open-Prescribing database for the same time periods, to assess overall opioid prescribing trends for all intervention and control practices [32]. We converted the monthly prescribing data into opioid prescribing monthly rate per 1,000 patients based on the 2017–2018 practice list size. We collected data from the 2017–2018 Public Health England National General Practice Profiles [33] for practice-level variables, comprising practice list size; female-to-male patient ratio; percentage of patients with long-term conditions, as a proxy for disease burden; and percentage of patients reporting a positive experience of their practice, as a marker of satisfaction with care. We used the percentage of patients in employment and practice-level Index of Multiple Deprivation (IMD) score as markers of deprivation. The IMD measures area deprivation and is determined for each patient on the list, where available, and then averaged over the practice. We used overall achievement in the clinical domain of the Quality and Outcomes Framework —a performance management system whereby primary care practices are remunerated according to achievement of targets—as a measure of overall quality of care [34].

We estimated intervention costs based on known costs (e.g., postage and data extraction fees) and time spent by staff (full-time equivalent salaries). Potential opioid prescription

savings were calculated based on national opioid prescription costs and trends for the West Yorkshire population. A formal economic analysis was not conducted.

### Data analysis

We used multilevel linear mixed-effects models (LMMs) for all outcomes. This was a 3-level model with a random intercept and random slope on month at the practice level, and a random intercept at the CCG level, with practice nested within CCG (S4 Text). The LMMs allowed the outcome to differ over time for each practice and accounted for correlations in outcomes over time within a practice and between practices within the same CCG area. A fixed-effect interaction term of intervention (control/intervention), the 3 intervention periods (pre-intervention/intervention/post-intervention), and month (July 2013 to December 2017) estimated the change in the outcomes over time across the 3 periods, and differences in change in outcomes between intervention and control practices, within a single model. We compared different structures of the covariance matrices (unstructured, independent, and identity) to assess which best accounted for autocorrelation. For all outcomes, the unstructured covariance (i.e., distinct variances and covariance) was the most appropriate, comparing both the Akaike information criterion (AIC) and Bayesian information criterion (BIC) values. Finally, each LMM included the predetermined practice characteristics as fixed effects, to assess whether any differences in the outcomes between the intervention and controls arms were due to practice differences. We checked that assumptions regarding autocorrelation—homoscedasticity of the residuals and normality of the residuals' distribution for LMMs—were not violated for all unadjusted and adjusted models. We confirmed that seasonality would not be an influence by reviewing changes in outcomes for each practice over time before developing models.

Sensitivity analysis (S5 Text) explored and confirmed the robustness of the modelling approaches, based on the main outcome adjusted LMM. We removed predicted values with residuals more than 2 or less than −2 to assess the impact of outliers; this made little difference to model estimates (S6 Text). Multicollinearity was not found for correlations between the practice characteristics ($\rho > 0.7$ and $p < 0.05$), and while some practice characteristics showed differences in rates of adults taking opioids at different levels of the practice characteristic (determined by including a 4-way interaction term with intervention, the 3 periods, and month), these differences did not change over time. Comparisons of AIC and BIC values for multilevel mixed-effects Poisson and negative binomial regression models and the adjusted LMM (all without CCG level due to convergence issues) for the main outcome indicated that the LMM was the most appropriate fit to the data (S6 Text).

We adhered to current reporting recommendations for ITS [35–37]. Our statistical analysis plan is provided (S7 Text).

### Ethical approval

The University of Leeds School of Medicine Research Ethics Committee provided ethical approval for the evaluation (MREC 17–042).

### Results

Intervention practices were similar to control practices but generally had larger list sizes, fewer patients with long-term conditions, and more deprived populations (Table 1). Before the intervention, the mean rate of adults prescribed opioids per 1,000 adults per month was 58.1 in intervention practices and 62.2 in control practices (Table 2). The number of patients at higher risk of long-term or stronger opioid prescribing who were prescribed opioids; the number of

**Table 1. Summary of practice characteristics.**

| Dataset and group | Number of practices | Median list size (IQR) | Mean percent female (95% CI) | Median percent positive patient experience (IQR)[a] | Mean percent with LTC (95% CI)[b] | Median percent QOF score (IQR)[c] | Mean percent IMD (95% CI)[d] |
|---|---|---|---|---|---|---|---|
| **CCG data** | | | | | | | |
| Control practices | 130 | 6,673 (4,102, 9,803) | 49.4 (49.0, 51.6) | 83.3 (76.5, 89.6) | 55.4 (54.0, 58.1) | 98.1 (96.1, 99.5) | 28.9 (26.5, 32.1) |
| Intervention practices | 313 | 7,550 (4,452, 10,540) | 49.2 (48.8, 51.4) | 83.8 (76.3, 89.7) | 51.0 (50.0, 53.5) | 98.1 (96.1, 99.4) | 30.3 (28.9, 33.0) |
| **OpenPrescribing data** | | | | | | | |
| Control practices | 264 | 7,131 (3,982, 9,878) | 51.5 (48.0, 55.3) | 86.4 (77.9, 91.6) | 54.9 (53.9, 57.4) | 98.6 (96.5, 99.8) | 25.1 (23.3, 28.0) |
| Intervention practices | 313 | 7,550 (4,452, 10,540) | 49.2 (48.8, 51.4) | 83.8 (76.3, 89.7) | 51.0 (50.0, 53.5) | 98.1 (96.1, 99.4) | 30.3 (28.9, 33.0) |

CCG, clinical commissioning group; GP, general practitioner; IMD, Index of Multiple Deprivation; LTC, long-term condition; QOF, Quality and Outcomes Framework.

[a]Results from GP patient survey question: 'Overall, how would you describe your experience of your GP practice'. The indicator value is the percentage of people who answered 'very good' or 'fairly good'.

[b]Results from GP patient survey question: 'Do you have any long-term physical or mental health conditions, disabilities or illnesses'. The indicator value is the percentage of people who answered 'Yes'.

[c]The percentage of all QOF points achieved across all domains as a proportion of all achievable points. (QOF is a financially incentivised quality improvement programme for all GP practices in England.)

[d]An overall measure of multiple deprivation experienced by people living in an area: the higher the score, the greater the deprivation.

patients prescribed NSAIDs, gabapentin, or pregabalin; and the number of patients referred to musculoskeletal services were similar between intervention and control.

For the primary outcome, the rate of any opioid prescribing rose across all practices during the pre-intervention period, increasing more in control than in intervention practices, with an adjusted change in rate of 0.36 (95% CI 0.27, 0.46) and 0.18 (95% CI 0.11, 0.25) adults prescribed opioids per 1,000 per month, respectively (Table 3). During the intervention period, the opioid prescribing rate rose by 0.53 per 1,000 per month (95% CI 0.29, 0.77) in control practices but fell in intervention practices by 0.12 per 1,000 per month (95% CI −0.30, −0.07),

**Table 2. Summary of opioid prescribing and other outcome-related characteristics at baseline for intervention and control practices.**

| Characteristic | Median (IQR) number of adults per 1,000 adults at baseline (2013 September) | | | |
|---|---|---|---|---|
| | **CCG data** | | **OpenPrescribing data** | |
| | **Control practices** | **Intervention practices** | **Control practices** | **Intervention practices** |
| Opioid prescription | 62.2 (49.7, 76.8) | 58.1 (44.9, 71.9) | 40.3 (30.6, 50.8) | 34.5 (25.7, 44.7) |
| Strong opioid prescription | 4.2 (2.8, 5.8) | 4.9 (3.2, 7.0) | | |
| Opioid prescription—patient >75 years | 108.1 (83.9, 138.0) | 119.5 (97.8, 143.7) | | |
| Anti-depressant prescription | 14.5 (10.0, 18.9) | 12.8 (9.1, 17.0) | | |
| Mental health diagnosis | 23.9 (17.6, 32.3) | 23.9 (17.7, 30.0) | | |
| Benzodiazepine prescription | 4.8 (3.3, 6.6) | 3.9 (2.1, 5.9) | | |
| Non-steroidal anti-inflammatory prescription | 27.0 (20.2, 34.1) | 27.9 (20.8, 40.5) | | |
| Gabapentin prescription | 8.0 (5.9, 10.8) | 6.4 (4.5, 8.8) | | |
| Pregabalin prescription | 5.3 (3.5, 7.2) | 5.2 (3.5, 7.2) | | |
| Musculoskeletal referral | 2.8 (1.9, 4.0) | 3.5 (2.4, 4.6) | | |

CCG, clinical commissioning group.

**Table 3. Mean number of adults prescribed opioid per 1,000 adults and mean change per month: multilevel linear model—electronic health record data and denominator.**

| Outcome and time period | Month | Mean (95% CI) number of adults prescribed opioid per 1,000 adults | | | Mean (95% CI) change per month, over the time period | | |
|---|---|---|---|---|---|---|---|
| | | Control (*n* = 130) | Intervention (*n* = 213) | Difference | Control (*n* = 130) | Intervention (*n* = 213) | Difference |
| **Adults prescribed opioid—unadjusted** | | | | | | | |
| Pre-intervention | 2013–09 | 57.3 (49.6, 64.9) | 57.0 (50.0, 63.9) | −0.3 (−10.6, 10.0) | 0.36 (0.27, 0.46) | 0.18 (0.11, 0.25) | −0.18 (−0.30, −0.07) |
| | 2016–03 | 68.2 (61.0, 75.4) | 62.4 (55.7, 69.0) | −5.8 (−15.6, 3.9) | | | |
| Intervention | 2016–04 | 63.9 (56.6, 71.1) | 63.7 (57.0, 70.3) | −0.2 (−10.1, 9.6) | 0.53 (0.29, 0.77) | −0.12 (−0.30, 0.07) | −0.65 (−0.95, −0.35) |
| | 2017–03 | 69.7 (62.5, 77.0) | 62.4 (55.7, 69.0) | −7.4 (−17.2, 2.5) | | | |
| Post-intervention | 2017–04 | 66.3 (59.0, 73.5) | 61.7 (55.0, 68.4) | −4.6 (−14.4, 5.3) | 0.22 (−0.03, 0.46) | −0.04 (−0.24, 0.15) | −0.26 (−0.57, 0.05) |
| | 2018–03 | 68.7 (61.3, 76.0) | 61.2 (54.5, 68.0) | −7.4 (−17.4, 2.6) | | | |
| **Adults prescribed opioid—adjusted**[a] | | | | | | | |
| Pre-intervention | 2013–09 | 55.0 (46.8, 63.2) | 58.2 (50.6, 65.9) | 3.2 (−8.0, 14.5) | 0.36 (0.27, 0.46) | 0.18 (0.11, 0.25) | −0.18 (−0.30, −0.07) |
| | 2016–03 | 65.9 (58.2, 73.7) | 63.6 (56.2, 71.0) | −2.3 (−13.1, 8.5) | | | |
| Intervention | 2016–04 | 61.7 (53.8, 69.6) | 64.9 (57.4, 72.4) | 3.2 (−7.7, 14.0) | 0.54 (0.29, 0.78) | −0.11 (−0.30, 0.08) | −0.65 (−0.96, −0.34) |
| | 2017–03 | 67.6 (59.7, 75.5) | 63.7 (56.2, 71.1) | −4.0 (−14.8, 6.9) | | | |
| Post-intervention | 2017–04 | 64.2 (56.3, 72.1) | 63.0 (55.5, 70.5) | −1.2 (−12.0, 9.7) | 0.21 (−0.03, 0.46) | −0.05 (−0.24, 0.15) | −0.26 (−0.57, 0.05) |
| | 2018–03 | 66.5 (58.6, 74.5) | 62.5 (55.0, 70.0) | −4.0 (−15.0, 7.0) | | | |
| **Adults prescribed strong opioid—adjusted**[a] | | | | | | | |
| Pre-intervention | 2013–09 | 3.9 (3.1, 4.7) | 4.9 (4.2, 5.6) | 1.0 (0.0, 2.1) | 0.04 (0.03, 0.05) | 0.04 (0.03, 0.05) | 0.002 (−0.01, 0.01) |
| | 2016–03 | 5.1 (4.4, 5.8) | 6.2 (5.5, 6.8) | 1.1 (0.1, 2.0) | | | |
| Intervention | 2016–04 | 4.4 (3.7, 5.2) | 6.2 (5.6, 6.8) | 1.8 (0.8, 2.7) | 0.01 (−0.01, 0.03) | −0.10 (−0.11, −0.08) | −0.11 (−0.13, −0.08) |
| | 2017–03 | 4.6 (3.9, 5.3) | 5.1 (4.5, 5.8) | 0.6 (−0.4, 1.5) | | | |
| Post-intervention | 2017–04 | 4.2 (3.5, 4.9) | 5.1 (4.5, 5.7) | 0.9 (−0.1, 1.8) | −0.03 (−0.05, −0.01) | −0.02 (−0.03, −0.003) | 0.01 (−0.01, 0.04) |
| | 2018–03 | 3.9 (3.2, 4.6) | 4.9 (4.2, 5.5) | 1.0 (0.1, 2.0) | | | |
| **Adults aged >75 years prescribed opioid—adjusted**[a] | | | | | | | |
| Pre-intervention | 2013–09 | 81.9 (64.4, 99.3) | 111.0 (95.8, 126.2) | 29.1 (5.9, 52.3) | 1.54 (1.33, 1.76) | 0.77 (0.60, 0.94) | −0.78 (−1.05, −0.50) |
| | 2016–03 | 128.2 (113.3, 143.1) | 134.1 (120.6, 147.6) | 5.9 (−14.3, 26.1) | | | |
| Intervention | 2016–04 | 106.5 (91.5, 121.4) | 137.4 (123.8, 151.0) | 30.9 (10.7, 51.2) | 1.82 (1.37, 2.27) | 0.06 (−0.28, 0.41) | −1.76 (−2.33, −1.19) |
| | 2017–03 | 126.5 (112.0, 141.0) | 138.1 (124.8, 151.4) | 11.6 (−8.2, 31.4) | | | |

(*Continued*)

**Table 3.** (Continued)

| Outcome and time period | Month | Mean (95% CI) number of adults prescribed opioid per 1,000 adults | | | Mean (95% CI) change per month, over the time period | | |
|---|---|---|---|---|---|---|---|
| | | Control (*n* = 130) | Intervention (*n* = 213) | Difference | Control (*n* = 130) | Intervention (*n* = 213) | Difference |
| Post-intervention | 2017–04 | 118.1 (103.6, 132.6) | 133.1 (119.8, 146.4) | 15.0 (−4.8, 34.7) | 0.72 (0.28, 1.17) | 0.09 (−0.26, 0.45) | −0.63 (−1.20, −0.06) |
| | 2018–03 | 126.1 (111.7, 140.5) | 134.1 (120.9, 147.3) | 8.1 (−11.6, 27.7) | | | |
| **Adults co-prescribed an antidepressant with opioid—adjusted[a]** | | | | | | | |
| Pre-intervention | 2013–09 | 11.7 (9.7, 13.8) | 11.7 (9.9, 13.5) | −0.03 (−2.8, 2.7) | 0.1 (0.07, 0.14) | 0.12 (0.09, 0.14) | 0.02 (−0.03, 0.06) |
| | 2016–03 | 14.8 (13.0, 16.7) | 15.3 (13.6, 17.0) | 0.4 (−2.1, 3.0) | | | |
| Intervention | 2016–04 | 14.2 (12.3, 16.2) | 16.2 (14.5, 18.0) | 2.0 (−0.6, 4.6) | 0.18 (0.09, 0.27) | −0.003 (−0.08, 0.07) | −0.18 (−0.30, −0.06) |
| | 2017–03 | 16.2 (14.3, 18.2) | 16.2 (14.4, 17.9) | −0.04 (−2.7, 2.6) | | | |
| Post-intervention | 2017–04 | 15.4 (13.5, 17.4) | 16.1 (14.3, 17.8) | 0.7 (−2.0, 3.3) | 0.13 (0.04, 0.23) | 0.04 (−0.03, 0.11) | −0.09 (−0.21, 0.03) |
| | 2018–03 | 16.9 (14.9, 18.9) | 16.5 (14.7, 18.3) | −0.4 (−3.1, 2.3) | | | |
| **Adults with a mental health diagnosis prescribed opioid—adjusted[a]** | | | | | | | |
| Pre-intervention | 2013–09 | 20.3 (16.0, 24.6) | 22.3 (18.2, 26.4) | 2.0 (−4.0, 7.9) | 0.2 (0.16, 0.23) | 0.14 (0.11, 0.16) | −0.06 (−0.11, −0.02) |
| | 2016–03 | 26.2 (22.0, 30.4) | 26.3 (22.3, 30.4) | 0.1 (−5.8, 5.9) | | | |
| Intervention | 2016–04 | 24.6 (20.4, 28.8) | 27.0 (22.9, 31.0) | 2.4 (−3.5, 8.2) | 0.27 (0.19, 0.35) | 0.03 (−0.03, 0.09) | −0.24 (−0.35, −0.14) |
| | 2017–03 | 27.6 (23.4, 31.8) | 27.3 (23.2, 31.3) | −0.3 (−6.2, 5.6) | | | |
| Post-intervention | 2017–04 | 26.2 (22.0, 30.4) | 27.0 (22.9, 31.1) | 0.8 (−5.1, 6.7) | 0.18 (0.10, 0.26) | 0.08 (0.02, 0.15) | −0.1 (−0.20, 0.004) |
| | 2018–03 | 28.2 (23.9, 32.5) | 27.9 (23.8, 32.0) | −0.3 (−6.2, 5.6) | | | |
| **Adults co-prescribed a benzodiazepine with opioid—adjusted[a]** | | | | | | | |
| Pre-intervention | 2013–09 | 5.9 (5.0, 6.7) | 4.6 (3.9, 5.2) | −1.3 (−2.4, −0.2) | −0.02 (−0.04, 0.01) | 0.02 (0.0002, 0.04) | 0.04 (0.006, 0.07) |
| | 2016–03 | 5.3 (4.6, 6.1) | 5.2 (4.6, 5.7) | −0.2 (−1.1, 0.7) | | | |
| Intervention | 2016–04 | 5.2 (4.4, 6.0) | 5.5 (4.8, 6.1) | 0.2 (−0.8, 1.3) | 0.05 (−0.02, 0.13) | −0.03 (−0.09, 0.03) | −0.09 (−0.19, 0.01) |
| | 2017–03 | 5.8 (5.0, 6.7) | 5.1 (4.4, 5.7) | −0.8 (−1.8, 0.3) | | | |
| Post-intervention | 2017–04 | 5.3 (4.5, 6.2) | 5.3 (4.7, 6.0) | 0.0 (−1.1, 1.1) | 0.02 (−0.06, 0.09) | −0.07 (−0.13, −0.01) | −0.09 (−0.19, 0.01) |
| | 2018–03 | 5.5 (4.6, 6.4) | 4.5 (3.8, 5.3) | −1.0 (−2.2, 0.2) | | | |
| **Adults prescribed a non-steroidal anti-inflammatory—adjusted[a]** | | | | | | | |
| Pre-intervention | 2013–09 | 35.2 (29.9, 40.6) | 40.8 (36.1, 45.4) | 5.5 (−1.6, 12.7) | −0.15 (−0.20, −0.10) | −0.08 (−0.12, −0.05) | 0.07 (0.005, 0.13) |
| | 2016–03 | 30.8 (25.8, 35.8) | 38.3 (33.8, 42.7) | 7.5 (0.8, 14.2) | | | |

(*Continued*)

**Table 3.** (Continued)

| Outcome and time period | Month | Mean (95% CI) number of adults prescribed opioid per 1,000 adults | | | Mean (95% CI) change per month, over the time period | | |
|---|---|---|---|---|---|---|---|
| | | Control (n = 130) | Intervention (n = 213) | Difference | Control (n = 130) | Intervention (n = 213) | Difference |
| Intervention | 2016–04 | 28.7 (23.6, 33.7) | 36.6 (32.2, 41.1) | 8.0 (1.2, 14.7) | −0.1 (−0.25, 0.05) | −0.35 (−0.47, −0.24) | −0.25 (−0.44, −0.06) |
| | 2017–03 | 27.6 (22.6, 32.6) | 32.8 (28.3, 37.2) | 5.2 (−1.5, 11.9) | | | |
| Post-intervention | 2017–04 | 26.3 (21.4, 31.3) | 30.8 (26.4, 35.2) | 4.5 (−2.2, 11.2) | −0.11 (−0.26, 0.03) | −0.16 (−0.28, −0.04) | −0.04 (−0.23, 0.15) |
| | 2018–03 | 25.1 (20.2, 30.0) | 29.1 (24.7, 33.5) | 4.0 (−2.7, 10.6) | | | |
| **Adults prescribed gabapentin—adjusted[a]** | | | | | | | |
| Pre-intervention | 2013–09 | 6.6 (4.4, 8.8) | 6.1 (4.3, 8.0) | −0.5 (−3.4, 2.5) | 0.07 (0.04, 0.11) | 0.11 (0.09, 0.14) | 0.04 (−0.005, 0.08) |
| | 2016–03 | 8.8 (6.6, 11.0) | 9.5 (7.6, 11.3) | 0.7 (−2.2, 3.5) | | | |
| Intervention | 2016–04 | 9.0 (6.7, 11.3) | 10.5 (8.6, 12.5) | 1.5 (−1.5, 4.5) | 0.1 (−0.04, 0.24) | −0.15 (−0.26, −0.04) | −0.25 (−0.42, −0.07) |
| | 2017–03 | 10.1 (7.8, 12.4) | 8.9 (7.0, 10.8) | −1.2 (−4.2, 1.8) | | | |
| Post-intervention | 2017–04 | 10.2 (7.9, 12.4) | 9.0 (7.0, 10.9) | −1.2 (−4.2, 1.8) | 0.03 (−0.11, 0.17) | −0.02 (−0.13, 0.09) | −0.05 (−0.23, 0.13) |
| | 2018–03 | 10.5 (8.2, 12.7) | 8.7 (6.8, 10.6) | −1.7 (−4.7, 1.2) | | | |
| **Adults prescribed pregabalin—adjusted[a]** | | | | | | | |
| Pre-intervention | 2013–09 | 8.4 (7.0, 9.7) | 5.0 (3.9, 6.1) | −3.4 (−5.1, −1.6) | −0.07 (−0.11, −0.04) | 0.06 (0.04, 0.09) | 0.14 (0.10, 0.18) |
| | 2016–03 | 6.1 (4.8, 7.5) | 6.8 (5.7, 7.9) | 0.7 (−1.0, 2.5) | | | |
| Intervention | 2016–04 | 5.6 (4.1, 7.1) | 6.3 (5.1, 7.5) | 0.7 (−1.2, 2.6) | −0.03 (−0.17, 0.10) | −0.02 (−0.13, 0.08) | 0.01 (−0.16, 0.18) |
| | 2017–03 | 5.3 (3.8, 6.8) | 6.0 (4.8, 7.2) | 0.8 (−1.2, 2.7) | | | |
| Post-intervention | 2017–04 | 3.4 (1.9, 4.9) | 5.2 (4.0, 6.4) | 1.8 (−0.1, 3.8) | 0.25 (0.11, 0.39) | 0.24 (0.14, 0.35) | −0.01 (−0.18, 0.17) |
| | 2018–03 | 6.1 (4.6, 7.6) | 7.9 (6.7, 9.1) | 1.8 (−0.2, 3.7) | | | |
| **Adults referred to musculoskeletal services—adjusted[a]** | | | | | | | |
| Pre-intervention | 2013–09 | 2.7 (2.0, 3.3) | 3.8 (3.2, 4.4) | 1.1 (0.3, 2.0) | 0.02 (0.009, 0.03) | 0.004 (−0.005, 0.01) | −0.02 (−0.03, −0.002) |
| | 2016–03 | 3.3 (2.8, 3.8) | 3.9 (3.4, 4.4) | 0.6 (−0.1, 1.3) | | | |
| Intervention | 2016–04 | 3.4 (2.8, 3.9) | 4.2 (3.7, 4.6) | 0.8 (0.1, 1.5) | −0.02 (−0.04, 0.01) | 0.003 (−0.01, 0.02) | 0.02 (−0.008, 0.05) |
| | 2017–03 | 3.2 (2.7, 3.7) | 4.2 (3.7, 4.7) | 1.0 (0.3, 1.7) | | | |
| Post-intervention | 2017–04 | 3.3 (2.8, 3.9) | 4.5 (4.0, 5.0) | 1.1 (0.4, 1.9) | −0.05 (−0.07, −0.02) | −0.05 (−0.07, −0.04) | −0.01 (−0.04, 0.02) |
| | 2018–03 | 2.8 (2.3, 3.4) | 3.9 (3.4, 4.4) | 1.1 (0.3, 1.8) | | | |

[a]Adjusted for percent female, Quality and Outcomes Framework score, percentage of patients reporting a positive experience of their practice, percentage of patients with long-term conditions, and Index of Multiple Deprivation.

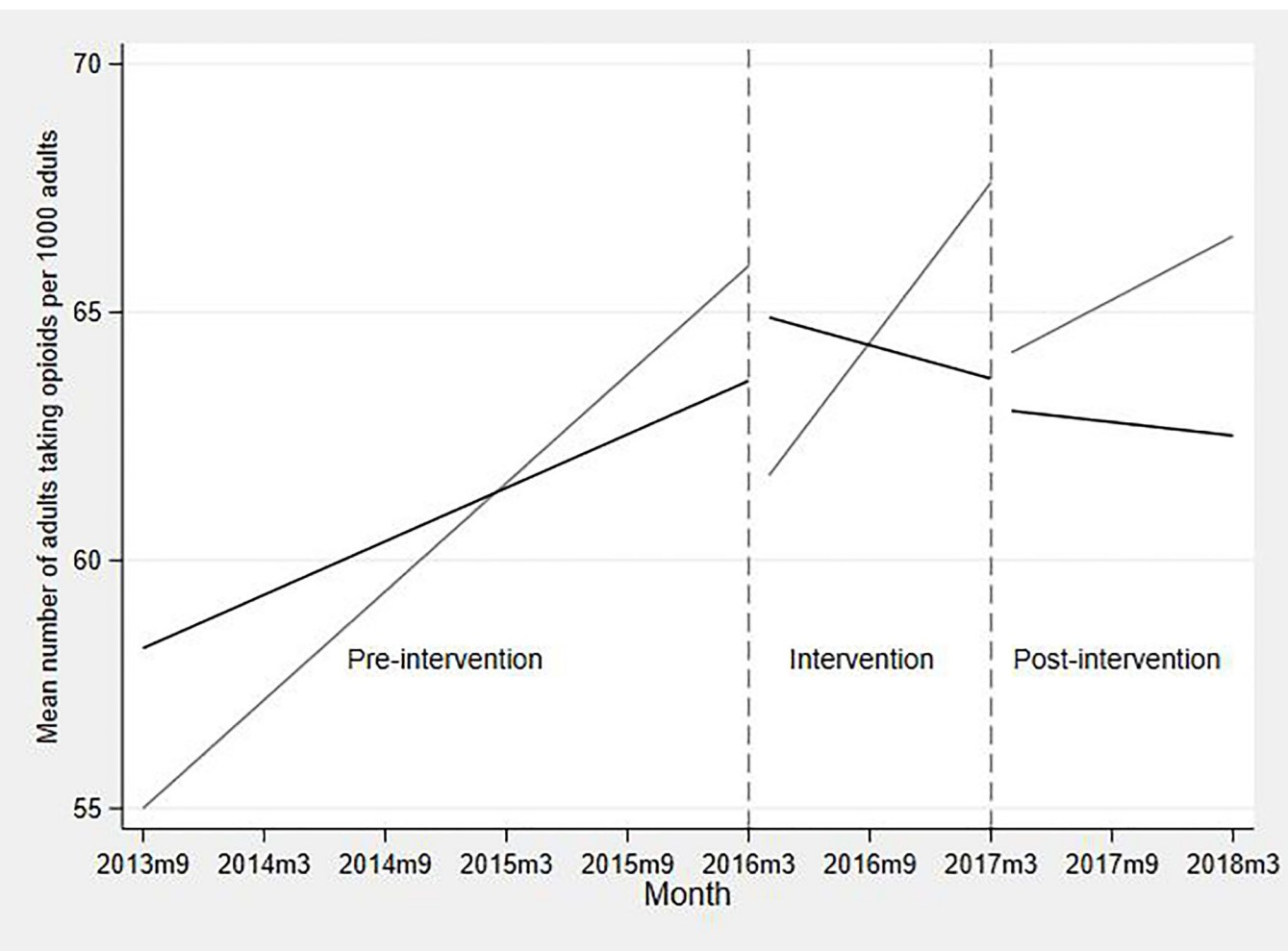

**Fig 1. Mean number of adults prescribed opioid per 1,000 adults: multilevel linear model estimates: Electronic health record data and denominator.**
Adjusted for percent female, Quality and Outcomes Framework score, percentage of patients reporting a positive experience of their practice, percentage of patients with long-term conditions, and Index of Multiple Deprivation. Black line = intervention practices; grey line = control practices.

a difference of 0.65 (95% CI −0.95, −0.35; Fig 1). Post-intervention, the opioid prescribing rates decreased in both groups, with a smaller difference in mean change per month between the control and intervention practices of 0.26 (95% CI −0.57, 0.05). By the final month of follow-up, there was a mean difference of 7.4 (95% CI −17.4, 2.6) per 1,000 adults prescribed opioids between control and intervention practices. We estimate that this corresponds to around 15,000 fewer adults prescribed opioids during the intervention year in our total intervention population of 1.9 million. Estimated intervention effects changed little after adjustment for practice characteristics, and therefore adjusted estimates are shown.

We observed trends generally favouring the intervention for groups at higher risk of long-term or stronger opioid prescribing. During the intervention, the rate of strong opioid prescribing decreased more in intervention than control practices (−0.11; 95% CI −0.13, −0.08), although rates in both groups similarly declined post-intervention. The rate of opioid prescribing in those aged 75 years and over decreased more in intervention practices than in control practices during the intervention period (−1.76; 95% CI −2.33, −1.19), with a sustained, if reduced, post-intervention difference (−0.63; 95% CI −1.20, −0.06). During the intervention

**Table 4. Mean number of prescriptions for opioids per 1,000 adults: Multilevel linear model—OpenPrescribing data, Public Health England National General Practice Profiles denominator.**

| Time period | Month | Mean (95% CI) number of prescriptions for opioids per 1,000 adults | | | Mean (95% CI) change per month, over the time period | | |
|---|---|---|---|---|---|---|---|
| | | Control (*n* = 264) | Intervention (*n* = 313) | Difference | Control (*n* = 264) | Intervention (*n* = 313) | Difference |
| Pre-intervention | 2013–09 | 38.7 (33.2, 44.2) | 33.5 (29.3, 37.6) | −5.3 (−12.1, 1.6) | 0.11 (0.06, 0.15) | 0.11 (0.09, 0.14) | 0.01 (−0.04, 0.06) |
| | 2016–03 | 41.9 (36.6, 47.2) | 36.9 (32.9, 40.9) | −5.0 (−11.7, 1.7) | | | |
| Intervention | 2016–04 | 43.3 (38.0, 48.6) | 37.9 (33.9, 41.9) | −5.4 (−12.1, 1.3) | 0.02 (−0.05, 0.10) | −0.08 (−0.13, −0.03) | −0.10 (−0.19, −0.01) |
| | 2017–03 | 43.5 (38.2, 48.8) | 37.0 (33.0, 41.1) | −6.5 (−13.2, 0.2) | | | |
| Post-intervention | 2017–04 | 43.7 (38.4, 49.0) | 37.1 (33.0, 41.1) | −6.7 (−13.4, 0.0) | −0.03 (−0.10, 0.05) | −0.05 (−0.10, 0.001) | −0.02 (−0.11, 0.07) |
| | 2018–03 | 43.4 (38.1, 48.8) | 36.5 (32.5, 40.6) | −6.9 (−13.6, −0.2) | | | |

CCG and practice levels, adjusted for percent female, Quality and Outcomes Framework score, percentage of patients reporting a positive experience of their practice, percentage of patients with long-term conditions, and Index of Multiple Deprivation.

period, rates of opioid prescribing fell more per month in intervention practices than in control practices in adults co-prescribed an antidepressant (−0.18; 95% CI −0.30, −0.06) and in adults with a mental health diagnosis (−0.24; 95% CI −0.35, −0.14), although post-intervention differences were not sustained. Rates of co-prescribed benzodiazepines did not differ significantly between intervention and control practices.

Regarding other analgesics, we observed declining pre-intervention trends for NSAID prescribing, with a larger decrease in intervention practices than control practices during the intervention (−0.35; 95% CI −0.47, −0.24) and both groups having similar post-intervention decreases. Rates of gabapentin prescribing decreased more in intervention practices than control practices during the intervention period (−0.25; 95% CI −0.42, −0.07), but this was not the case for pregabalin prescribing (0.01; 95% CI −0.16, 0.18). We observed no differences in rates of musculoskeletal referrals between intervention and control practices during the intervention period (0.02; 95% CI −0.008, 0.05) or after (0.01; 95% CI −0.04, 0.02).

Using publicly available data for total opioid prescriptions, we observed rising pre-intervention trends for both groups, a small decline during the intervention in intervention practices (−0.1; 95% CI −0.19, −0.01), and fairly static post-intervention rates in both groups (Table 4).

The results of a simple (uncontrolled) ITS of intervention practices mirrored those of the controlled ITS (Table 5). This provides greater confidence that any association between the intervention and the effect is likely to be causal, and provides evidence that the control practices did not experience some other event [35].

We estimated that the feedback intervention cost approximately US$66,000 to deliver, including US$52,000 in staff costs, US$3,200 in data extraction fees, and US$5,200 in stationary costs. Nationally, opioid prescription costs rose by approximately US$26,000 per 100,000 population during the intervention year. The reduction in opioid prescribing equated to around US$1,155,000 savings across intervention CCGs. The intervention gave overall cost savings of US$1,000,000 once all costs were accounted for.

## Discussion

We observed that repeated evidence- and theory-informed comparative feedback reversed a rising trend of opioid prescribing in primary care, with sustained, if attenuated, effects. We have therefore demonstrated a successful, scalable strategy to reduce population-level opioid prescribing. The feedback intervention had a modest effect, with a difference of 0.65 fewer adults prescribed any opioid per 1,000 per month in intervention practices compared to

**Table 5. Mean number of adults prescribed opioid per 1,000 adults: Multilevel linear model—electronic health record data and denominator, intervention only model (n = 313).**

| Time period | Month | Mean (95% CI) number of adults prescribed opioid per 1,000 adults—adjusted | Mean (95% CI) change per month, over the timeframe |
|---|---|---|---|
| Pre-intervention | 2013–09 | 57.1 (54.3, 70.6) | 0.18 (0.10, 0.25) |
| | 2016–03 | 62.4 (55.6, 72.1) | |
| Intervention | 2016–04 | 63.7 (55.5, 71.9) | −0.12 (−0.32, 0.08) |
| | 2017–03 | 62.4 (54.3, 70.6) | |
| Post-intervention | 2017–04 | 61.8 (53.6, 70.0) | −0.04 (−0.25, 0.16) |
| | 2018–03 | 61.3 (53.1, 69.6) | |

control practices. However, at a population level, there were substantially fewer patients taking prescribed opioid medications.

The number of patients prescribed strong opioids fell during the intervention, although at a slower rate than the number of patients prescribed any opioid, possibly reflecting a longer de-prescribing process than for weaker opioids, given the need for gradual reductions to limit withdrawal symptoms. The intervention also had sustained effects for patients in targeted high-risk groups, including adults with coded mental health diagnoses and those co-prescribed antidepressants. The greatest effect was in adults aged 75 years and older, with a greater reduction in intervention practices than control practices of almost 1.8 adults aged 75 years and older prescribed opioids per 1,000 per month. This is important given the heightened risks of premature mortality, associated falls, and unplanned hospital admissions in this population [38,39].

Contrary to expectations, we observed reductions in wider analgesic prescribing not specifically targeted by feedback, specifically of NSAIDs and gabapentin, and no increases in referrals to musculoskeletal services. This provides some reassurance that the intervention had few rebound effects on wider service utilisation and costs. Indeed, it may have prompted primary care physicians to think differently about the value of prescribing analgesics in chronic non-cancer pain, and to prefer self-management options.

Prescribing data from publicly available sources [32] confirm that the intervention changed the underlying trend of rising opioid prescriptions, although it levelled off rather than fell. The smaller effect in this dataset is likely due to additional 'noise' in these data, which include prescriptions for cancer pain and drug dependency, especially as primary care physicians are encouraged to prescribe stronger opioids earlier and longer for palliative care [40].

There is a growing evidence base on the value of provider- and system-level interventions to reduce opioid use in adults with chronic non-cancer pain [19–21,41,42]. We provide evidence for a relatively efficient and scalable population strategy to address prescribing of both weaker and stronger opioids. The widespread use of EHR systems means that primary care prescribing data can be used to both drive and monitor change at a relatively low cost [43–45]. Our estimated costs suggest this intervention is relatively efficient given potential savings in projected opioid prescription costs.

Our intervention incorporated a range of evidence- and expert-informed suggestions to improve the effectiveness of feedback, such as providing repeated feedback with comparators to reinforce desired behaviour, recommending specific actions, and ensuring credibility of information [30]. However, the success of our strategy may also have depended upon

contextual factors, specifically the timing and nature of the targeted clinical behaviour [46]. The intervention occurred during a period when primary care physicians were becoming increasingly aware of an opioid prescription problem and recognised a need for action. Feedback, used alone or with other interventions, may not be effective in changing all types of clinical behaviour [29]; opioid prescribing represents a relatively discrete behaviour that is reasonably within physician control [30].

We highlight 5 limitations. First, our study took place in a single region, potentially limiting generalisability to the rest of the UK and other healthcare systems. However, primary care physicians internationally report similar types of challenges in managing opioid prescribing [47], and performance feedback has been shown to work in many settings [22]. As only 1 out of 317 practices declined participation, selection bias is unlikely. We also demonstrated effects in a population with relatively high levels of socioeconomic deprivation, a factor that is associated with higher levels of opioid prescribing [48].

Second, routinely collected data are prone to coding errors. Such errors are less likely for prescribing data, but our use of 'ever coded' diagnoses may have overestimated current diagnoses, especially cancer and drug dependence. Some practices may have responded to feedback by re-categorising patients as drug dependent, thereby taking them out of the denominator and inflating intervention effects. However, we observed similar patterns of reductions in OpenPrescribing data. Sensitivity analysis showed that 'extreme' values, possibly due to coding errors, did not affect model estimates. Furthermore, the modelling approaches accounted for missing data within practices.

Third, the quasi-experimental design cannot fully account for concurrent interventions. Our previous publication showing the rise in prescribing in this area may have alerted practices to rising opioid prescribing [16]. Media attention to the North American 'opioid crisis' during the intervention period may also have influenced prescribing behaviour. Media coverage of the scale of UK opioid prescribing began towards the end of the intervention period and is unlikely to significantly account for observed changes in opioid prescribing [49,50]. Use of control practices [26] and a simple ITS analysis provides greater confidence that any association between intervention and effect is likely to be causal.

Fourth, this study did not specifically examine the acceptability of the feedback reports and whether or how they were used by general practices. This will be addressed in a separate process evaluation.

Fifth, we cannot be certain whether reductions in opioid prescribing were always clinically appropriate as we did not assess individual patient clinical indications and outcomes. The absence of any increases in prescribing of other potentially harmful analgesics and in referrals suggests that the intervention did not generate increased demand.

Patients have strong expectations for prescription pain relief, making reductions in prescribing challenging if they are perceived as undermining therapeutic relationships and patient satisfaction. Strategies to bring about significant improvements in healthcare delivery are unlikely to succeed if they fail to address multiple barriers and enablers. Addressing the rise of opioid prescribing and its legacy is likely to require sustained, coordinated efforts across all levels of healthcare systems that target organisational, clinical, and patient behaviours [51]. Performance feedback offers one approach that can be coupled with complementary educational campaigns and decision support to change physician prescribing habits and patient expectations [22]. We welcome further research to determine whether our findings can be replicated in other healthcare systems. There are further opportunities to evaluate and enhance feedback effectiveness, ideally involving head-to-head comparisons of different ways of delivering feedback within randomised designs [52].

## Conclusions

We observed that an evidence- and theory-informed feedback intervention reversed rising opioid prescribing trends in a primary care setting. Effects decreased following cessation of the feedback, which may need to be sustained for maximum long-term impact. We observed no concurrent increases in prescribing of other analgesics or demand for musculoskeletal services. Feedback therefore offers a scalable approach to reduce population-level opioid prescribing.

## Supporting information

**S1 Text. TIDieR checklist for the Campaign to Reduce Opioid Prescribing intervention, incorporating the reporting and design elements of audit and feedback intervention recommendations.**
(PDF)

**S2 Text. Sample practice report.** The baselayer of the map used in this file is from https://commons.wikimedia.org/wiki/File:England_Clinical_Commissioning_Group_(CCG)_Map_(Labelled).svg.
(PDF)

**S3 Text. Primary outcome search terms for The Phoenix Partnership SystmOne electronic health record system.**
(PDF)

**S4 Text. Multilevel linear mixed-effects model.**
(PDF)

**S5 Text. Sensitivity analysis.**
(PDF)

**S6 Text. Mean number of adults prescribed opioid per 1,000 adults: Multilevel linear model—electronic health record data and denominator, residuals ±2 removed.**
(PDF)

**S7 Text. Statistical analysis plan.**
(PDF)

## Acknowledgments

We would like to thank Mohammed Imran at West Yorkshire Research and Development for his role in data collection.

## Author Contributions

**Conceptualization:** Sarah L. Alderson, Paul Carder, Robbie Foy.

**Data curation:** Sarah L. Alderson, Paul Carder, Stella Johnson.

**Formal analysis:** Tracey M. Farragher.

**Funding acquisition:** Sarah L. Alderson.

**Investigation:** Sarah L. Alderson.

**Methodology:** Tracey M. Farragher, Thomas A. Willis, Robbie Foy.

**Project administration:** Sarah L. Alderson.

**Supervision:** Robbie Foy.

**Writing – original draft:** Sarah L. Alderson.

**Writing – review & editing:** Sarah L. Alderson, Tracey M. Farragher, Thomas A. Willis, Paul Carder, Stella Johnson, Robbie Foy.

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
