## [Editor Report · Decision Letter 0]

23 Nov 2020

Dear Dr Alderson, 

Thank you for submitting your manuscript entitled "The effects of an evidence and theory-informed feedback intervention on opioid prescribing for non-cancer pain in primary care: a controlled interrupted time series analysis" for consideration by PLOS Medicine.

Your manuscript has now been evaluated by the PLOS Medicine editorial staff and I am writing to let you know that we would like to send your submission out for external peer review.

Kind regards,

Artur A. Arikainen,

Associate Editor

PLOS Medicine

---

## [Decision Letter · Decision Letter 1]

16 Jan 2021

Dear Dr. Alderson,

Thank you very much for submitting your manuscript "The effects of an evidence and theory-informed feedback intervention on opioid prescribing for non-cancer pain in primary care: a controlled interrupted time series analysis" (PMEDICINE-D-20-05335R1) for consideration at PLOS Medicine. 

Your paper was evaluated by a senior editor and discussed among all the editors here. It was also discussed with an academic editor with relevant expertise, and sent to independent reviewers, including a statistical reviewer (r#2). The reviews are appended at the bottom of this email and any accompanying reviewer attachments can be seen via the link below:

[LINK]

In light of these reviews, I am afraid that we will not be able to accept the manuscript for publication in the journal in its current form, but we would like to consider a revised version that addresses the reviewers' and editors' comments. Obviously we cannot make any decision about publication until we have seen the revised manuscript and your response, and we plan to seek re-review by one or more of the reviewers. 

We expect to receive your revised manuscript by Feb 08 2021 11:59PM. Please email us (plosmedicine@plos.org) if you have any questions or concerns.

We look forward to receiving your revised manuscript. 

Sincerely,

Emma Veitch, PhD

PLOS Medicine

On behalf of Clare Stone, PhD, Acting Chief Editor, 

PLOS Medicine

plosmedicine.org

*In the last sentence of the Abstract Methods and Findings section, please include a note about any key limitation(s) of the study's methodology.

*In the abstract, effects and confidence intervals are currently a bit hard to read and could be reformatted so the reader can follow these better eg:

"The number of adults prescribed any opioid rose pre-intervention in both intervention and control practices BY 0.18 (95% CI; 0.11, 0.25) and 0.36 (95% CI; 0.27, 0.46) per 1,000 adults per month respectively. During the intervention period, prescribing fell in intervention practices (change -0.11; 95% CI -0.30, -0.08) and continued rising in control practices (change 0.54; 95% CI 0.29, 0.78) with a difference of -0.65 per 1000 patients (95% CI -0.96, -0.34)"

*At this stage, we ask that you include a short, non-technical Author Summary of your research to make findings accessible to a wide audience that includes both scientists and non-scientists. The Author Summary should immediately follow the Abstract in your revised manuscript. This text is subject to editorial change and should be distinct from the scientific abstract. Please see our author guidelines for more information: https://journals.plos.org/plosmedicine/s/revising-your-manuscript#loc-author-summary

*On page 9 of the paper, currently the authors note a sensitivity analysis that isn't included with the paper (below); the journal prefers to ensure that all claims made in the paper are backed up with analyses presented and there are no page/length limits; we'd recommend that the sensitivity analysis noted below is included, with supplementary materials if needed:

"Sensitivity analysis (not shown) explored and confirmed the robustness of the modelling approaches, based on the main outcome adjusted LMM"

*On page 12, the paper references some analyses on acceptability and value of the feedback intervention (below). If the PLOS Medicine paper is accepted then it won't be possible to include a reference to a manuscript under review, and the best outcome is if the acceptability/value paper is accepted for publication by this stage (and then it can be cited and included in the reference list as "in press" - or with a full citation/DOI). Please consider this when you come to resubmit the revision, it's also possible that the way this article is cited can be updated later, if the PLOS Medicine paper gets to the stage of provisional acceptance and materials are being prepared for publication. If the acceptability/value paper isn't accepted by this point, then the authors would need to be prepared to remove any reference/citation to it. 

"Our process evaluation suggests the acceptability and perceived value of credible, tailored feedback targeting issues of emerging or established concern (manuscript under review)"

*Did your study have a prospective protocol or analysis plan? Please state this (either way) early in the Methods section.

Comments from the reviewers:

Reviewer #1: 

There is an urgent need for evidence-based interventions to reduce inappropriate opioid prescribing at both population and individual patient level and this work certainly has potential to contribute to the former category. The supplementary information regarding the intervention characteristics (TIDiER checklist) and example feedback were very helpful. The overall finding of reduced opioid prescribing in practices receiving a fairly straightforward and low cost feedback intervention is very welcome. However, 3 years have elapsed since the intervention period, during which time there have been changes in prescriber awareness and behaviour that may impact the findings and I have some reservations about how the study is reported which are summarised below. 

Introduction 

Paragraph 1

Whilst it is true that the US has an opioid crisis - the scale of this may have peaked around 2016, based on CDC data from 2017- 18. Reference 7 is particularly old & I suggest using that to highlight that in the last 15-20 years an opioid crisis developed - in the US & other western countries before adding more up to date information from the US, UK and elsewhere that suggests the rise in prescribing may have peaked but hasn't fallen so much as may have been expected. I recommend highlighting other opioid-related harms - not just dependency e.g. overdose & falls/ fractures

Paragraph 2

Line 61-62 - there is a lack of evidence for all kinds of interventions - why highlight only psychological?

I don't think ref 16 is particularly well summarised in line 62-63 - is the struggle with managing chronic pain & the lack of effective medicines rather than with opioids per se?

Line 65 - reference 17 seems to be wrong

Methods

Study design and setting

Line 78 - whilst accepting this is worded for an international audience, it seems odd not to mention General Practice or GP practices when discussing UK primary care.

I suspect controlled ITS is not a study design that many readers will be very familiar with and I think a more detailed description of that and the rationale for using it would more useful to most readers than some of the detail given about UK CCGs etc., which is to some extent repeated in the results section.

I'm aware that there are not such clear reporting guidelines for studies using ITS design as for RCTs - but I think there are areas where more detail is needed. 

How did the authors decide on the sample size - i.e. number of practices included - and what is the justification for the substantial difference in numbers of intervention and control practices?

Discussion and conclusions

The study findings are reported as though they arise form an RCT rather than a quasi-experimental study design and whilst the shortcomings of the design is acknowledged to some extent, the authors conclude that "Repeated evidence and theory informed comparative feedback reversed a rising trend of opioid prescribing in primary care" but does this study design really permit such conclusions about causation?

Review by a methodologist and statistician is recommended - for a more expert opinion on these aspects than I can give.

Reviewer #2: 

Thanks for the opportunity to review your manuscript. My role is as a statistical reviewer so my queries are focused on study design, data, and analysis.

This study uses a multi-level interrupted time series analysis with control sites to test the effect of an intervention intended to reduce prescription of opioid analgesics. . I have put overall queries first, and then followed by questions related to a specific section of the manuscript with a page/line reference.

Was there any information available in the study from the practices about whether the intervention reports were used by physicians (or others in the practices?), and what they thought of the information (i.e. acceptability)?

P6. L86. Is the same database (electronic prescribing data) used throughout the different CCGs? 

P6. L86. How were the areas/practices that received the intervention decided? Was this random allocation or deliberate? The limitations of the study mention the quasi-experimental design but not specifically whether there were systematic differences between the intervention and control areas and whether these could lead to differing temporal changes.

P6. L87. To clarify, these extra practices were in the Yorkshire + Humber region, but not in the West Yorkshire area, and not selected from the five CCGs that provide control data? 

P6. L87. Is the data source for these extra practices the same as from the other sites (ie. Just a difference means for data provision)? 

P6.L87. Do patients always use the same practice? Is there any information on how many patients would move practices throughout the study period?

P6, L98. Were there other services available to patients in this area to replace opioid prescriptions? i.e. specialist pain management? 

P8. L127. Is it possible to derive total morphine equivalent dose from the prescription data? Or is lowering any opioid prescribing the main aim of the intervention, not lowering the dose or substituting a less strong agent?

P8. L144. This is a reasonably complicated model and I think it would be beneficial to include this in an appendix as a formula. Was a 'shift' parameter at the beginning of the intervention period used or just change over time?

P8. L164. For which outcomes was the Poisson model used, and which for the Neg Bin? How was the decision to use these made?

P11, L219. Were these stratified analyses done by restricting the patients in the analysis, or by including an interaction (e.g. age x control/int variable)? It isn't detailed in the methods how this was done. Was the effect in over 75s tested directly or is this a comparison of stratified analyses? 

Figure 1. Are these the estimated rates from the mixed models? Is it possible to also show the crude rates as well as the estimates in each month of the study? I would also consider modifying the x-axis, to make this easier to read (e.g. maybe have each year in labels and ticks for each month/quarter?)

Reviewer #3: 

This article, "The effects of an evidence and theory-informed feedback intervention on opioid prescribing for non-cancer pain in primary care: a controlled interrupted time series analysis", has significant merit in that it is reporting a large scale, low intensity intervention and its population level impact. Because the focus was on population exposure to opioids, the primary outcome, appropriately chosen was any adult prescribed an opioid. This is essentially how many people are exposed to opioids and not necessarily any impact on people already getting opioids, particularly those with risky opioids use.

By section:

Abstract- well written, clearly stated.

Introduction:

First paragraph- 

Line 55- North America is experiencing an opioid crisis but rising opioid mortality is related more to heroin and synthetic opioids than prescription opioids. The prescription opioid crisis occurred in the 2000s-2010 after which heroin (2010) and then fentanyl (2014) took over as leading causes of opioid overdose death. Important not to promulgate misinformation about the contribution of prescription opioids to the current crisis. 

Line 58- Opioids are likely to be no more effective than non-opioid pain medication (see Krebs SPACE trial published in JAMA in 2019), which doesn't mean they are of limited effectiveness. It is important to be clear that opioids do help some people but clearly have potential for great harm and don't help everyone. 

Second paragraph;

Contrary to line 60's assertion, a number of approaches have been tested to decrease opioid prescribing. There is a literature on ECHO to reduce opioid prescribing e.g. https://www.ncbi.nlm.nih.gov/pmc/articles/PMC6420488/; also, implementation of prescription drug monitoring programs (vast literature showing decreased in population based prescribing associated with PDMP), clinical intervention to improve guideline adherent prescribing using academic detailing and nurse care managers (https://pubmed.ncbi.nlm.nih.gov/28715535/, https://pubmed.ncbi.nlm.nih.gov/32697847/) and academic detailing alone (plentiful literature on this). 

Methods:

Study Design:

It might be useful to define "lists" in this first paragraph since this is used later in the manuscript, particularly in the tables, and it would be useful to understand how a list relates to the description of the practices for those in North America not familiar with the system in the UK. 

Intervention:

Line 93: It would be useful to name the evidence and theory that drive the intervention. The manuscript is diminished by lack of a published protocol that could have outlined this in detail. 

Data Sources and Outcomes:

The primary outcome is appropriate for this intervention. It wasn't clear why the particular high risk groups were targeted and what the goal was- For example, patients prescribed anti-depressants and opioids. Was the denominator people prescribed anti-depressants and the goal to decrease opioids or was it to ensure that people prescribed opioids also be prescribed anti-depressants for concomitant depression and for analgesic impact? Similarly, the strong vs. weak opioid analysis wasn't clearly justified although it more intuitive. Was it based on Morphine Equivalent doses? If so, the reporting on the data might include that, since someone one a weak opioid might be prescribed high amounts and someone on a stronger opioid might be prescribed just a few tablets. In the US, hydrocodone is one of the most commonly prescribed opioids but this was not listed here. Perhaps that is due to a different name in the UK or that it isn't on the formulary. Lastly, excluding individuals with drug dependence may exclude people prescribed opioids and then later diagnosed with drug dependence as a result of the prescription. 

The use of gabapentin and NSAID was useful to include as a comparator. 

Data Analysis- 

A statistician should comment on these analyses, but to a non-statistician, this looks complete and well detailed. 

The discussion reports on cost estimates, but there is no description where those calculations come from. 

See comment below on results/tables

Results:

The reporting of the results in the body of the manuscript seem clear but don't seem to match up with the values in the tables, which were a bit confusing. For example, the results in table 3 showed two values- the beginning of the period and the end of the period. But in some cases, the numbers didn't make sense. For example, number of adults prescribed opioids per 1000 control group 2016m3 was 68.2 but the next month 2016m4 was 63.9. Why was there such a disparity month to month? The same with the figure- it looks like it drops precipitously in a month period. Why wasn't this displayed for a rolling average? It might not be right to display this way, but in Larochelle, 2016 Annals of Internal Medicine, appendix figure 2, we showed a daily average before and after an index event, which was steadier over time, reflecting the reality of opioid prescribing. This article might have something similar but using an 8 week rolling average of adults per 1000 given opioids. The fact that the data is displayed this way makes me question whether all months of data were used, and if so, why weren't they incorporated into tables and figures?

It wasn't clear why the authors did an uncontrolled Interrupted time series of the intervention practices alone and it doesn't augment any of the findings or discussion in the manuscript. This is superfluous data analysis. 

Cost estimates should go in Results section along with details of how they were calculated prior to the discussion. 

Discussion: 

Paragraph 1- it would be useful to show the estimates of the population impact earlier, in the results

Paragraph 2- line 216-17 Since there was no analysis of MME, just the type of opioid, it isn't at all clear why there would need to be a gradual reduction to limit withdrawal symptoms. Most people on moderate doses would not experience withdrawal. 

Paragraph 3- the reduction of non-opioid medication and referrals may have had positive financial impact, but it may be that practitioners are leaving patients without any treatment for pain. There was no evidence of self-management options in the data reported. Furthermore, it generally takes longer to work with patients on self-management than a prescription medication so it is hard to know what actually occurred. Best to leave out speculation and do some future studies examining what did happen for patients with pain. 

Paragraph 5- see comment above about presenting cost data in the discussion without presenting first in the results. That being said, this has important implications for the intervention's impact on the population and health service. 

Paragraph 10 (lines 273-276) The lack of information on what actually happened in the clinical interaction with the patient is one of the most important limitations of the study. The generalizability and coding errors were expected as part of the study but would be balanced by the large size and reach (generalizability) as well as balance between intervention and control (coding errors). What we don't know is how the patients fared with the intervention. The lack of referrals and non-opioid medications is not necessarily reassuring because it could indicate that the pain was not addressed at all. 

Tables:

Table 1: 

It would be important to define some of the headings in expanded footnotes below, including:

List

Patient Experience- what does median patient experience mean- what measure is used and what is the scale? 

LTC- what are included in these?

QoF- What does % score mean?

IMD- What does % IMD indicate? 

Table 2: 

Baseline characteristics- are these a 12 month average? One month average? The time frame needs to be defined. 

CCG should be defined in footnote of table

Table 3: 

See critique above- 

Also, this table is very busy with lots of detail. It would be useful to simplify, if possible. Perhaps show mean numbers for the pre-intervention, intervention and post-intervention periods rather than the multiple data points. 

Table 4: 

Similar critique as for Table 3- could simplify it

In summary, this is an important intervention study with powerful population level results. It could use improvement in each of the main section of the paper and in the display of data. 

Jane Liebschutz, MD MPH

University of Pittsburgh

[LINK]

---

## [Decision Letter · Decision Letter 2]

4 Mar 2021

Dear Dr. Alderson,

Thank you very much for re-submitting your manuscript "The effects of an evidence and theory-informed feedback intervention on opioid prescribing for non-cancer pain in primary care: a controlled interrupted time series analysis" (PMEDICINE-D-20-05335R2) for review by PLOS Medicine.

I have discussed the paper with my colleagues and the academic editor and it was also seen again by the below reviewers. I am pleased to say that provided the remaining editorial and production issues are dealt with we are planning to accept the paper for publication in the journal.

[LINK]

We look forward to receiving the revised manuscript by Mar 11 2021 11:59PM.   

Sincerely,

Dr Raffaella Bosurgi, 

Executive Editor 

PLOS Medicine

plosmedicine.org

Requests from Editors:

Comments from Reviewers:

Reviewer #1: The authors have taken on board reviewer feedback. As a result, the revised manuscript is much improved and reads well - thank you! I have no further suggestions

Reviewer #2: Thanks for the opportunity to see your revised manuscript. Overall I think that the changes in the version resolve all of my queries from the first version I reviewed. The data sources section is clearer and the inclusion of the SAP was helpful for me as well.

The supplementary material was a useful addition. I can now follow the analysis with the equation for the main LMM model displayed. The Model fit table (S2) supports the use of the LMM and the results of the sensitivity analysis are similar during the pre and intervention periods. The post-intervention change over time seems to shift in the intervention areas, becoming positive whereas before it declined (although it's still lower than the control areas). The numbering of these tables may need to be adjusted as Table S1 follows Table S2 in the appendix. 

I think the information you provided about how the study groups were formed (request from staff at practices that eventually received the intervention, and then surrounding area) should be presented in the methods. 

This is a nice study - clearly lots work went into this and it has resulted in a good manuscript.

Reviewer #3: Overall, this is a much improved manuscript. It reads more clearly and was very responsive to the reviewer comments. I was not familiar with CITS and it was helpful to get the explanations provided in the response to reviewers, particularly about interpretation of the tables. 

There are a few things remaining that would be useful to clarify.

It was not clear to me until reading the response to the reviewers that this study took advantage of a natural experiment to examine the impact of a public health/health policy-driven intervention to address a rise in opioid prescribing in Leeds and Bradford by conducting a quasi-experimental analysis. This might be of interest to the readers, which also puts into context the the entire project. While statisticians are likely to understand that a controlled interrupted time series is quasi-experimental, many readers who are interested in clinical interventions for opioids prescribing may not understand this. Thus, I suggest that it be clarified in the methods section that the intervention was implemented in targeted areas and the control sites were identified to match the region and other characteristics of the intervention site. And it might be useful to add the term quasi-experimental in the abstract to alert readers unfamiliar with controlled interrupted time series. In my mind, the quasi experimental design is an important limitation as these sites were aware on some level that they were receiving this intervention because of poor performance. While the analysis makes a convincing argument that the intervention was impactful, there may be other forces at play. This should be added to the limitations. 

There is some confusion about the number of control practices. In the abstract, it says 130 control practices. In the Author Summary it says 187 practices. In the methods, on line 130, it says 187 practices provided control data, and on line 132 it was 134 practices were added as controls. 

In my prior review, I may not have been clear about my question about anti-depressants. If I understand the response to reviewers, anti-depressants were used as a proxy for mental illness in addition to analysis of diagnoses of mental illness. It might be helpful to be explicit about that in the manuscript. Other studies of pain control have used anti-depressants as a specific intervention for chronic pain, so higher antidepressant may be interpreted as a therapeutic intervention, not a proxy for mental illness. 

Tables: table 1 much improved- it would be useful to put the explanations below the table in order that the columns appear in the table. Patient experience is currently at the bottom despite being the 5th column. 

Minor typo- line 278 used the Pound rather than Dollar sign.

[LINK]

---

## [Decision Letter · Decision Letter 3]

3 Sep 2021

Dear Dr Alderson, 

On behalf of my colleagues and the Academic Editor, Dr. Zirui Song, I am pleased to inform you that we have agreed to publish your manuscript "The effects of an evidence and theory-informed feedback intervention on opioid prescribing for non-cancer pain in primary care: a controlled interrupted time series analysis" (PMEDICINE-D-20-05335R3) in PLOS Medicine.

PRESS

Sincerely, 

Beryne Odeny 

Associate Editor 

PLOS Medicine